# Visual impairment and refractive error in school children in Bhutan: The findings from the Bhutan School Sight Survey (BSSS 2019)

Indra Prasad Sharma[1]*, Nor Tshering Lepcha[1], Tshering Lhamo[1], Leon B. Ellwein[2], Gopal Prasad Pokharel[3], Taraprasad Das[4], Yuddha Dhoj Sapkota[5], Tandin Dorji[6], Sonam Peldon[6]

1 Gyalyum Kesang Choeden Wangchuck National Eye Center, JDW National Referral Hospital, Thimphu, Bhutan, 2 National Eye Institute, National Institute of Health, Bethesda, Maryland, United States of America, 3 Nepal Eye Hospital, Tripureshwor, Kathmandu, Nepal, 4 International Agency for Prevention of Blindness, South East Asia Regional Office, Hyderabad, India, 5 International Agency for Prevention of Blindness, South East Asia, Kathmandu, Nepal, 6 Healthcare Diagnostic Division, Department of Medical Services, Ministry of Health, Thimphu, Bhutan

* indrapsharma@gmail.com

**Data Availability Statement:** All relevant data are within the paper and its Supporting Information files.

## Abstract

### Purpose

To estimate the nationwide prevalence of visual impairment and associated refractive error in school children in Bhutan.

### Methods

The sample of this prospective cross-sectional national survey comprised of randomly selected classes in levels IV-IX (age 10 to 15 years) from schools throughout Bhutan. The examination included measurement of visual acuity (VA), evaluation of ocular motility, refraction under cycloplegia, examination of the external eye, media and fundus. The principal cause of impairment was determined for eyes with uncorrected VA ≤6/12. The main outcome measures were distance VA and cycloplegic refractive error.

### Results

With a sampling frame of 1967 class-based clusters from 190 schools, 160 classes in 103 schools were randomly selected; 4985 (98.5%) of 5060 enumerated children were examined. The prevalence of uncorrected, presenting, and best-corrected visual impairment (VA≤6/12) in the better eye was 14.5%, 12.8%, and 0.34%, respectively. Refractive error was the principal cause (94.2%) of impaired vision and 88% of children who could achieve VA ≥6/9 with best correction were without necessary spectacles. The prevalence of myopia (≤ -0.5 D) was 6.64% and was associated with female gender (P = 0.004), urban schooling (P = 0.002), and greater parental education (P<0.001). The prevalence of hyperopia (≥ +2.0 D) was 2.17% and was significantly associated with lower class-level (P = 0.033), and female gender (P = 0.025). The overall prevalence of astigmatism (≥ 0.75 D) was 9.75%.

**Funding:** This work was supported by Lions Club International Foundation (LCIF) SightFirst Research Grant (SFP 2053/UND). The funders had no role in study design, data collection and analysis, decision to publish, or preparation of the manuscript.

**Competing interests:** The authors have declared that no competing interests exist.

## Conclusions

Reduced vision because of uncorrected refractive error is a public health problem among school-age children in Bhutan. Effective school eye health strategies are needed to eliminate this easily treatable cause of visual impairment.

## Introduction

The Kingdom of Bhutan (henceforth 'Bhutan'), with a population of approximately 800,000, is small, landlocked country with some of the most rugged high mountains in the southern slopes of the Himalayas. Guided by a developmental philosophy of Gross National Happiness (GNH), Bhutan places a high priority to health and education sectors as its constitutional mandate. The state provides free education to all children of school-going age up to tenth standard and free access to basic public health services to its citizens [1].

Uncorrected refractive error is the leading cause of visual impairment; 48.9% globally and 62.9% in South Asia (Bhutan is part of this region) as estimated by the Vision Loss Expert Group (VLEG) [2]. Recent studies document a wide global variation in the prevalence of refractive error and predict that without control interventions the prevalence of myopia will significantly increase globally, affecting nearly 5 billion people by 2050 [3]. Considering its high prevalence and significant socioeconomic impact, refractive error has gained priority as a public health challenge, particularly in children [4].

Vision 2020: The Right to Sight initiative has mandated its member countries, including Bhutan, to generate baseline data on refractive error. To address the widespread need with comparable data, WHO has developed the Refractive Error Study in Children (RESC) protocol with uniform examination methods and definitions [5]. The RESC methodology was previously used for large-scale population-based surveys in Brazil, Chile, China, India, Malaysia, Nepal, and South Africa [6–16].

Though Bhutan falls in the high-risk region for refractive error, long-term strategic plans and policies could not be developed due to lack of adequate data on refractive error. The National Housing and Population Survey (2017) of Bhutan revealed that 2.9% of the population lives with visual disability, the most common amongst all disabilities [17]. The Bhutan School Sight Survey (BSSS), the first in Bhutan and carried out by the Primary Eye Care Program (PECP) in the Ministry of Health, was designed to assess the prevalence of visual impairment and refractive error in school-age children on a national level using the RESC protocol.

## Methodology

### Study population

The target population for this countrywide survey was children in class levels IV through IX (usually 10–16 years of age) attending lower, middle, and higher secondary schools. The sampling frame was constructed by identifying individual classes (clusters) within each of the IV-IX class levels at 190 (182 public and 8 private) schools in all the 20 districts (Dzongkhags) and four municipalities (Thromdes); 64 lower secondary schools, 78 middle secondary schools, and 48 higher secondary schools. The resulting sampling frame comprised of an estimated 1967 classes (clusters), with cluster-based sampling, stratified by class-level to ensure a reasonably uniform distribution of age across the study population.

## Sample size

The required sample size was based on estimating a 15% prevalence of refractive error with a 20% error (15% ± 3.0%). With simple random sampling, 544 children were required for each class-level stratum. Assuming 5% non-participation the required sample size increased to 573, and accounting for a 50% increase because of inefficiencies associated with the cluster sampling design further increased the required sample size to 860. Anticipating an average 35 students per class, the selection of 25 classes was required for each of the six class-level strata, which corresponds to a total study sample of 150 classes and expected 5250 children.

## Informed consent and approval

Enumeration of student's name, age, gender, parent/guardian name and education was completed prior to the examination. Study information including the objectives of the study and details regarding the eye examination was explained, and written informed consent was obtained from the parents or guardians for each child enrolled in this study.

Human subject approval for the original RESC study protocol was obtained from the World Health Organization (WHO) Secretariat Committee on Research Involving Human Subjects.

The Research Ethics Board of Health (REBH), Ministry of Health, Government of Bhutan approved the implementation of the study and the Ministry of Education provided the administrative approvals. The study adhered to the tenets of the Declaration of Helsinki on human subjects participation.

## Field operations

The field personnel were trained to perform clinical procedures, complete data forms, and quality assurance. The data manager was trained in the use of data entry, data cleaning and statistical analysis software. Pilot exercises were conducted to ensure familiarity with all aspects of the protocol and measurement methods in a field setting. Monitoring and supervision were conducted throughout the period of fieldwork to circumvent unintentional protocol deviations. The fieldwork was carried out between March 2019 and June 2019.

## Ophthalmic examinations

The clinical team comprised of one senior optometrist and two ophthalmic assistants. All clinical examinations were conducted in temporary stations set-up in each school. Distance visual acuity was measured at six meters using a retro-illuminated visual acuity (VA) chart with tumbling-E optotypes (Appasamy, India) beginning with the top line (6/24) and advancing through the 6/18, 6/12, 6/9 and 6/6 lines. Tumbling optotypes were used so that the children do not memorize the letters. If at any level the child failed to recognize at least four optotypes, the line immediately above the failed line was retested, until successful. If the top line was not read correctly, the child was tested with a Snellen visual acuity test chart that contained additional lines (6/60 and 6/36). The lowest line read successfully was assigned as the visual acuity for the tested eye. VA was measured in the right eye first, followed by the left eye, each time occluding the fellow eye, and without and with spectacles, if any.

Binocular motor function was assessed with cover-uncover test at both 0.5 (near) and 6.0 (distance) meters. Corneal light reflex was used to quantify the degree of tropia. A magnifying binocular loupe and a flashlight were used to examine the anterior segment (eyelid, conjunctiva, cornea, iris, and pupil). The pupils were dilated with 1% cyclopentolate in children with unaided VA 6/12 or worse in either eye. Pupils were considered fully dilated if 6 mm or

greater, and cycloplegia was considered complete if the pupillary light reflex was absent. Cycloplegic refraction was performed with a Plusoptix A12R autorefractometer (Plusoptix GmbH, Nuernberg, Germany) with daily calibration. Autorefraction was followed by subjective refraction and measurement of best corrected VA. The optometrist performed a dilated fundus examination using a direct/indirect ophthalmoscope. The principal cause of visual impairment (VA $\leq$ 6/12) was assigned for each eye; refractive error was routinely assigned as the cause for eyes improving to $\geq$6/9 with subjective refractive correction. Children with vision $\leq$6/12 in one or both eyes improving with refractive correction were provided with the prescription glasses, and others were referred.

## Data management and analysis

Class (cluster) enumeration and individual examination data forms were reviewed for accuracy and completeness prior to computerized data entry in Microsoft Access. Data ranges, frequency distributions, and consistency among related measurements were checked with data analysis programs.

The prevalence of visual impairment was calculated from the uncorrected, presenting and best corrected VA. Visual acuity categories were defined as: normal/near-normal vision ($\geq$6/9 in both eyes); unilateral visual impairment ($\geq$6/9 in one eye only); mild impairment ($\leq$6/12 to $\geq$6/18 in the better eye); moderate impairment ($\leq$6/24 to $\geq$6/48 in the better eye); and severe visual impairment ($\leq$6/60 in both eyes).

Myopia was defined as spherical equivalent (SE) refractive error of at least -0.50 diopters (D) and hyperopia as +2.00 D or more. Visually impaired children were considered myopic if one or both eyes were myopic and hyperopic if one or both eyes were hyperopic so long as neither eye was myopic. Astigmatism was defined as cylinder values of 0.75 D or more.

The prevalence of ametropia was calculated in children with visual impairment in at least one eye (i.e., the prevalence of visual impairment *with* hyperopia and the prevalence of visual impairment *with* myopia). Children with normal or near-normal vision in both eyes were not refracted, and thus, were not included in the hyperopia and myopia prevalence calculations.

The association between myopia and hyperopia with age and class-level, gender, school location (urban vs rural), student type (day scholar vs boarder) and parental education was explored using multiple logistic regression. Parental education (parents with the highest degree of schooling) was categorized as none (no formal schooling), primary, low/middle secondary, higher secondary, and tertiary (college/university).

Statistical analysis was performed using Stata Statistical Software (Release 12.0, StataCorp, College Station, Texas). Confidence intervals and P values (significant at the p $<$0.05 level) for prevalence estimates and regression models were calculated with adjustment for clustering effects associated with the sampling design.

## Results

### Study sample

Based on a random selection, 162 classes from 109 schools were identified; data of children from 160 classes in 103 participating schools were completed and analyzed. Four randomly selected private schools were removed from the study sample because of their small number and two public schools could not be reached because of poor weather-related connectivity.

A total of 5060 children were enumerated among which 4985 (98.5%) were examined. Distributions of age, gender, class-level, school location, student type, and parental education of both enumerated and examined populations are shown in S1 Table. Although the number of classes in each class-level was essentially similar (generally 26 or 27), the average number of

children examined per class increased from 28 for class-level IV to 34 for class-level IX; The higher number of students in upper classes was because of a large number of sections in lower classes. Accordingly, the total number of children examined ranged from 755 in class-level IV to 918 in class-level IX. The average ages (± standard deviation) across classes V through IX were 10.42 (±1.05), 11.27 (±1.07), 12.33 (±1.11), 13.45 (±1.27), 14.58 (±1.30), and 15.57 (±1.22) years, respectively.

## Visual acuity

Visual acuity findings are presented in Table 1. Uncorrected normal/near-normal visual acuity ($\geq$6/9) in at least one eye was found in 85.5% (n = 4261) children. Vision impairment ($\leq$ 6/12) in both eyes was detected in 14.5% (n = 724) children and 1.08% (n = 54) children had severe visual impairment ($\leq$6/60).

Overall, 3.27% (n = 163) children were wearing spectacles during the examination (Table 1). The study detected that only 19.9% (144 of 724) children with uncorrected visual acuity impairment in both eyes were wearing spectacles at the time of examination and 11.9% (86 of 724) children were improving to normal/near-normal vision in at least one eye with the current spectacles. In a multiple logistic regression (with uncorrected visual acuity, age, gender, school location and parental education as covariates) female children (odds ratio, OR 1.92 [95% Confidence interval, CI: 1.33–2.77]; P<0.001) and children of parents with higher education (OR 1.12 [95% CI: 0.96–1.31]; P = 0.154) were more likely to be wearing glasses.

Of the 638 children (12.8% of those examined) presenting with bilateral visual impairment, 97.3% (n = 621) could achieve normal/near-normal vision in at least one eye with best correction (Table 1). For the 17 remaining children with bilateral visual impairment, none were severely impaired. Uncorrectable visual impairment was caused by amblyopia in 10 children, with 8 not meeting the defined criteria, and by unexplained or undetermined causes in 7 children.

## Pupillary dilation and cycloplegia

Cycloplegic dilation was not done/ not achieved in 7 of the 782 visually impaired right eyes and 6 of the 786 visually impaired left eyes because of phthisis, corneal opacity, and an

**Table 1. Distribution of uncorrected, presenting and best-corrected visual acuity.**

| VA Category | Uncorrected VA | Wearing Glasses | Presenting VA | Best-Corrected VA |
|---|---|---|---|---|
| | No. (%; 95% CI) | No. (%) * | No. (%; 95% CI) | No. (%; 95% CI) |
| Normal/near normal[†] | 4141 (83.1; 81.5–84.6) | 2 (0.05) | 4214 (84.5; 83.1–86.0) | 4911 (98.5; 98.2–98.9) |
| Unilateral impairment[‡] | 120 (2.41; 1.91–2.90) | 17 (14.2) | 133 (2.67; 2.16–3.17) | 57 (1.14; 0.84–1.44) |
| Mild impairment in better eye[‖] | 522 (10.5; 9.27–11.7) | 55 (10.5) | 517 (10.4; 9.19–11.5) | 13 (0.26; 0.12–0.40) |
| Moderate impairment in better eye[§] | 148 (2.97; 2.48–3.45) | 54 (36.5) | 102 (2.05; 1.60–2.49) | 4 (0.08; 0.01–0.16) |
| Severe impairment in both eyes[#] | 54 (1.08; 0.77–1.40) | 35 (64.8) | 19 (0.38; 0.20–0.56) | 0 (0.00) |
| All | 4985 (100.0) | 163 (3.27) | 4985 (100.0) | 4985 (100.0) |

CI, confidence interval; VA, visual acuity.

* Percent of the number within each VA category based on uncorrected vision.

[†] $\geq$6/9 in both eyes.

[‡] $\geq$6/9 in one eye only.

[‖] $\leq$6/12 to $\geq$6/18.

[§] $\leq$6/24 to $\geq$6/48.

[#] $\leq$6/60 in both eyes.

irregular pupil. Pupillary dilation of at least 6 mm and cycloplegia (absence of light reflex) was achieved in 78.4% of right eyes with another 20.7% having cycloplegia but without full dilation for a total of 99.1% with cycloplegia; it was 77.1%, 22.1%, and 99.2%, respectively, for the left eyes.

## Refractive error

Refractive error measurements were available for 769 of the 782 visually impaired right eyes and for 780 of the 786 visually impaired left eyes. Refractive error was based on autorefraction except for 2 right eyes and 1 left eye where subjective refraction was used, as the autorefraction data were not available. Although refraction measurements were not available for 19 visually impaired eyes, none of the 844 children with unilateral or bilateral visual impairment were without at least one measurement.

Overall, the prevalence of visual impairment associated with hyperopia was 2.17%, with substantial variation within age (class-level), gender and parental education subgroups (Table 2). The overall prevalence of myopia was 6.64%, increasing from 2.78% at age 10 (class IV) to 10.8% at age 15 years (class IX).

Multiple logistic regression was used to quantify the association of hyperopia and myopia with age and class-level, gender, school location, student type and parental education (Table 3). Separate age and class-level models were used to address significant pairwise interactions between age and class-level covariates. Hyperopia was marginally associated with female gender in the age model; higher class-level, and greater parental education was significant in the class-level model. Myopia was associated with elderly children, and higher class-level, female gender, urban schooling and greater parental education in both models.

## Astigmatism

Astigmatism $\geq$0.75 D in *either* eye was present in 9.75% (n = 486) children and 2.4% (n = 119) children had astigmatism $\geq$2.00 D (S2 Table). In multiple logistic regression modeling with age and gender as covariates, astigmatism was marginally associated with female gender (OR 1.17 [95% CI: 0.96–1.41]; P = 0.120), but not with age (P = 0.822).

## Binocular motor function

Thirty six (0.72%) children had tropia; 27 (0.54%) children at near fixation and 30 (0.60%) at distance fixation. Exotropia was more common; 59.3% for near (81.5% < $15^0$) and 73.3% for distance (76.7% < $15^0$).

## Causes of visual impairment

Refractive error was the principal cause of visual impairment in this cohort of children. The majority (91.2%; n = 770 of 844) of children with visual impairment in one or both eyes attained normal or near normal acuity in both eyes with best correction (Table 1). Twenty-seven additional children had correctable refractive error in one eye with an uncorrectable cause in the fellow eye, for a total of 797 (94.4%) with refractive error as the principal cause of impairment in at least one eye (Table 4).

Most prominent among other causes of visual impairment was amblyopia, which was diagnosed as the principal cause by the optometrist in 43 (5.09%) children, but only 19 (2.25%) of these met the predefined criteria. Those meeting the criteria included 15 with anisometropia $\geq$2.00 SE diopters (including 6 also with hyperopia $\geq$6.00 SE diopters and 2 also with exotropia), 2 with only hyperopia, and another 2 with only esotropia. Other causes of visual

**Table 2. Prevalence of hyperopic and myopic visual impairment by age, gender, class-level, school location, student type and parental education.**

| | Children with Visual Impairment | | | Children without Visual Impairment | Total Number Examined |
|---|---|---|---|---|---|
| | Hyperopia* | Myopia† | Emmetropia | | |
| | No. (%; 95% CI) | No. (%; 95% CI) | No. (%) | No. (%) | |
| Age (yrs) | | | | | |
| 9–10 | 20 (3.21; 1.75–4.67) | 26 (4.17; 2.31–6.04) | 45 (7.22) | 532 (85.4) | 623 |
| 11–12 | 38 (2.49; 1.67–3.31) | 85 (5.57; 4.31–6.83) | 118 (7.73) | 1285 (84.2) | 1526 |
| 13–14 | 29 (1.95; 1.19–2.72) | 103 (6.94; 5.61–8.27) | 120 (8.09) | 1232 (83.0) | 1484 |
| 15–16 | 13 (1.25; 0.50–1.99) | 88 (8.43; 6.48–10.4) | 98 (9.40) | 844 (80.9) | 1043 |
| 17–18 | 8 (2.59; 0.64–4.53) | 29 (9.38; 6.11–12.7) | 24 (7.77) | 248 (80.3) | 309 |
| Gender | | | | | |
| Male | 42 (1.73; 1.19–2.28) | 138 (5.57; 4.73–6.70) | 192 (7.95) | 2044 (84.6) | 2416 |
| Female | 66 (2.57; 1.91–3.23) | 193 (7.51; 6.31–8.72) | 213 (8.29) | 2097 (81.6) | 2569 |
| Class-Level | | | | | |
| IV | 28 (3.71; 2.14–5.28) | 21 (2.78; 1.21–4.36) | 60 (7.95) | 646 (85.6) | 755 |
| V | 11 (1.36; 0.59–2.13) | 45 (5.56; 3.66–7.47) | 41 (5.07) | 712 (88.0) | 809 |
| VI | 27 (3.38; 2.11–4.64) | 51 (6.38; 4.54–8.21) | 77 (9.63) | 645 (80.6) | 800 |
| VII | 19 (2.25; 1.00–3.49) | 57 (6.74; 4.63–8.85) | 67 (7.92) | 703 (83.1) | 846 |
| VIII | 9 (1.05; 0.22–1.88) | 58 (6.77; 4.86–8.68) | 71 (8.28) | 719 (83.9) | 857 |
| IX | 14 (1.52; 0.63–2.42) | 99 (10.8; 8.34–13.2) | 89 (9.68) | 717 (78.0) | 918 |
| School Location | | | | | |
| Rural | 55 (2.07; 1.42–2.72) | 142 (5.35; 4.28–6.43) | 238 (8.97) | 2218 (83.6) | 2653 |
| Urban | 53 (2.27; 1.62–2.93) | 189 (8.10; 6.84–9.37) | 167 (7.16) | 1923 (82.5) | 2332 |
| Student Type | | | | | |
| Day | 80 (2.25; 1.71–2.79) | 242 (6.80; 5.81–7.79) | 287 (8.06) | 2950 (82.9) | 3559 |
| Boarder | 28 (1.96; 1.13–2.79) | 89 (6.24; 4.72–7.76) | 118 (8.27) | 1191 (83.5) | 1426 |
| Parental Education | | | | | |
| None | 35 (1.81; 1.17–2.45) | 113 (5.84; 4.80–6.88) | 154 (7.96) | 1632 (84.4) | 1934 |
| Primary | 18 (1.62; 0.83–2.43) | 59 (5.34; 3.74–6.94) | 84 (7.60) | 944 (85.4) | 1105 |
| Low/Mid Secondary | 19 (2.35; 1.07–3.64) | 50 (6.20; 4.35–8.04) | 76 (9.42) | 662 (82.0) | 807 |
| Higher Secondary | 21 (3.25; 1.79–4.70) | 57 (8.81; 6.60–11.0) | 49 (7.57) | 520 (80.4) | 647 |
| Tertiary | 13 (2.95; 1.50–4.41) | 48 (10.9; 8.25–13.6) | 35 (7.95) | 344 (78.2) | 440 |
| Missing Information | 2 (3.85) | 4 (7.69) | 7 (13.5) | 39 (75.0) | 52 |
| All | 108 (2.17; 1.71–2.62) | 331 (6.64; 5.82–7.46) | 405 (8.12) | 4141 (83.1) | 4985 |

CI = confidence interval.

* Spherical equivalent of +2.00 diopters (D) or more in either eye.

† Spherical equivalent of -0.50 D or more in either eye.

impairment included corneal opacity/scar (n = 5) and cataract and retinal disorder (n = 1 each). Twenty-one eyes of 15 children had visual impairment related to nystagmus, phthisis, trauma/injury and optic atrophy, including 4 eyes with no light perception. We could not identify the cause of impaired vision in 10 eyes of 9 children.

## Discussion

This study, using a robust RESC protocol, was designed to estimates of the prevalence of visual impairment and associated refractive error in school children in Bhutan. This is the first nationwide survey to provide a population-based, internationally comparable data and representative information of practical value in planning, implementation and monitoring of

**Table 3. Logistic regression models for hyperopic and myopic visual impairment with age (or class-level), gender, school location, student type and parental education as covariates.**

| | Hyperopia* | | Myopia† | |
|---|---|---|---|---|
| | OR (95% CI); *P*-value | | OR (95% CI); *P*-Value | |
| | **Model with Age** | **Model with Class Level** | **Model with Age** | **Model with Class Level** |
| Older Age | 0.92 (0.81–1.05); *P* = 0.220 | ———— | 1.23 (1.16–1.30); *P*<0.001 | ———— |
| Higher Class-Level | ———— | 0.86 (0.74–0.99); *P* = 0.033 | ———— | 1.32 (1.22–1.43); *P*<0.001 |
| Female Gender | 1.54 (1.04–2.30); *P* = 0.033 | 1.58 (1.06–2.34); *P* = 0.025 | 1.43 (1.12–1.83); *P* = 0.004 | 1.36 (1.07–1.73); *P* = 0.013 |
| Urban School | 1.03 (0.65–1.64); *P* = 0.900 | 1.05 (0.66–1.69); *P* = 0.824 | 1.62 (1.20–2.18); *P* = 0.002 | 1.56 (1.16–2.10); *P* = 0.004 |
| Boarder Student | 1.17 (0.68–2.03); *P* = 0.562 | 1.27 (0.73–2.21); *P* = 0.397 | 1.00 (0.70–1.43); *P* = 0.984 | 0.98 (0.69–1.38); *P* = 0.897 |
| Greater Parental Education | 1.14 (0.96–1.36); *P* = 0.127 | 1.15 (0.98–1.36); *P* = 0.083 | 1.29 (1.17–1.42); *P*<0.001 | 1.23 (1.12–1.34); *P*<0.001 |

OR, odds ratio; CI, confidence interval.

* Spherical equivalent of +2.00 diopters (D) or more in either eye.

† Spherical equivalent of -0.50 D or more in either eye.

refractive error services. It showed that a majority of children (88%) who could achieve normal or near normal visions with refractive correction were without the necessary spectacles. This information is of public health importance and is similar with the ones seen in other parts of the world [6–16].

Because of the similarity of the geographical terrain and ethnicity, the comparison of our data with those published from Nepal, a neighboring country, is informative. The school eye-health studies in Nepal have been done in public and private schools in different districts and geographic regions, in street children and children of parents in the upper middle class economy [18–24]. These studies have confirmed that uncorrected refractive error is the most common eye disorder in children, with the prevalence of myopia higher than hyperopia. The prevalence of myopia was less in rural and Sub Himalayan Nepal [6, 18], more prevalent in children in private schools [20, 21], in children with higher parental economic status than the

**Table 4. Causes of uncorrected visual acuity (UCVA) of 6/12 or worse.**

| Principal Cause of Impairment | Eyes with UCVA ≤6/12 | | Children with UCVA ≤6/12 in One or Both Eyes* | |
|---|---|---|---|---|
| | **Right Eye** | **Left Eye** | **Either Eye** | **% Prevalence in the Population** |
| | **No. (%)** | **No. (%)** | **No. (%)** | |
| Refractive Error† | 733 (93.7) | 744 (94.7) | 797 (94.4) | 16.0 |
| Amblyopia Criteria‡ | 30 (3.84) | 23 (2.93) | 43 (5.09) | 0.86 |
| Corneal Opacity/Scar | 3 (0.38) | 2 (0.25) | 5 (0.59) | 0.10 |
| Cataract | 0 | 1 (0.13) | 1 (0.12) | 0.02 |
| Retinal Disorder | 0 | 1 (0.13) | 1 (0.12) | 0.02 |
| Other Causes | 11 (1.41) | 10 (1.27) | 15 (1.78) | 0.30 |
| Unexplained Cause | 5 (0.64) | 5 (0.64) | 9 (1.07) | 0.18 |
| Any Cause | 782 (100.0) | 786 (100.0) | 844 (100.0) | 16.9 |

* Children with visual acuity ≤6/12 in both eyes may represent two different causes of reduced vision; a different cause for each eye. Accordingly, the 844 children with any cause of impairment is exceeded by the total number with specific causes (871). Similarly, the any cause of prevalence is exceeded by the total for cause-specific prevalence.

† Refractive error was assigned as the cause of reduced vision for all eyes correcting to ≥6/9 with subjective refraction, even if other contributing pathology was present.

‡ Defined tropia, anisometropia, and hyperopia criteria for amblyopia were met in 21 eyes of 19 children.

street children [15, 24] and more often higher in girls [15]. The variation in the prevalence of myopia was attributed to stress in life, including classroom work and parental pressure.

The overall prevalence of visual impairment (uncorrected visual acuity ≤6/12 in one or both eyes) was 16.9%, with refractive error accounting for 94.4% of this impairment. Amblyopia, related to refractive error, was the principal cause in at least one eye in 5.09% of children. Other relatively infrequent causes included corneal opacity/scars, cataract, retinal disorders, nystagmus, phthisis, trauma/injury, and unexplainable causes.

The prevalence of hyperopia and myopia was 2.17% and 6.64%, respectively. These estimates of hyperopia and myopia prevalence pertain only to children with visual impairment in one or both eyes. Accordingly, to the extent that some children with normal or near normal vision may have been ametropic, our data are likely to underestimate the actual prevalence; representing a limitation of the RESC protocol.

The findings from the current study indicate that myopia prevalence, which increases with class-level (age), is higher among female children, children attending urban schools, and children of parents with greater levels of education. These differences underscore the significance of environmental influences on myopia development, possibly relating to a combination of more time spent on near work in urban school children with educated parents (associated with a greater emphasis on schooling and academic performance), along with less time on outdoor activities outside school hours. Near work in children has been shown to be associated with myopia development, and outdoor activity has been shown to be negatively associated [25, 26]. In general, boys have more outdoor activities than girls; we suspect it could, at least in part, be the reason for myopia prevalence between the two genders [26]. Barriers to use of corrective spectacles include parental awareness of the vision problem, attitudes regarding the need for spectacles, spectacles cost, cosmetic appearance, and the belief that wearing glasses may cause progression of refractive error [27].

The finding that the visual impairment with presenting VA is much worse than that with best corrected VA, and not much improved over uncorrected VA, indicate that many children were in need of appropriate refractive correction. (Only 161 of the 844 presenting with visual impairment were wearing glasses, with only 73 of these improving to normal vision. With best corrected refraction for all children, including those not well corrected with existing glasses, another 697 would achieve normal vision.) Additionally, the finding of a low prevalence (0.34%) of visual impairment with best corrected VA indicates a relatively low prevalence of causes other than refractive error.

The strengths of the survey included a large, randomly selected, sample of school children throughout Bhutan which ensured representativeness, the use of the RESC study protocol with standardized measurement methods and definitions for valid and direct comparisons with other RESC studies, and high examination response rates precluded self-selection bias. This study is representative of visual acuity screening and refractive error as a cause of visual impairment across the entire country.

It is apparent that regular eye and vision screening are important in the schools of Bhutan, along with parental education. Consecutively, it is equally important to improve the quality of optometric services and provision of affordable spectacles to address this huge unmet need for refractive correction. Follow-up studies to assess the changing trends in refractive errors and compliance to spectacle wear is necessary. Bhutan could also consider education policy to moderate indoor schooling time and encourage increased out-door activities to reduce risks of developing or progressing myopia. Long-term strategic plans, policies and programs have to be planned and implemented to have a sustainable impact.

## Supporting information

**S1 Table. Distribution of enumerated and examined populations by age, gender, class-level, school location, student type and parental education.**
(DOCX)

**S2 Table. Prevalence of astigmatism by age and gender.**
(DOCX)

**S1 File. Rawdata file.**
(XLS)

## Acknowledgments

The authors acknowledge Essilor International for providing free spectacles for children and Mission for Vision (MFV), India for donating equipment for the study. We thank the International Agency for Prevention of Blindness (IAPB), South East Asia for technical support. We also acknowledge the Royal Government of Bhutan for overall guidance and support, study enumerators for their work in data collection, and school officials for their assistance and cooperation in conducting this study.

## Author Contributions

**Conceptualization:** Indra Prasad Sharma, Nor Tshering Lepcha, Leon B. Ellwein, Gopal Prasad Pokharel, Taraprasad Das, Yuddha Dhoj Sapkota, Tandin Dorji.

**Data curation:** Indra Prasad Sharma, Nor Tshering Lepcha, Tshering Lhamo, Gopal Prasad Pokharel, Yuddha Dhoj Sapkota.

**Formal analysis:** Indra Prasad Sharma, Tshering Lhamo, Leon B. Ellwein, Gopal Prasad Pokharel, Yuddha Dhoj Sapkota.

**Funding acquisition:** Indra Prasad Sharma, Taraprasad Das, Yuddha Dhoj Sapkota.

**Investigation:** Indra Prasad Sharma, Nor Tshering Lepcha, Tshering Lhamo, Gopal Prasad Pokharel, Taraprasad Das, Sonam Peldon.

**Methodology:** Indra Prasad Sharma, Nor Tshering Lepcha, Tshering Lhamo, Leon B. Ellwein, Gopal Prasad Pokharel, Yuddha Dhoj Sapkota, Tandin Dorji, Sonam Peldon.

**Project administration:** Taraprasad Das, Tandin Dorji, Sonam Peldon.

**Resources:** Leon B. Ellwein, Taraprasad Das, Yuddha Dhoj Sapkota, Tandin Dorji, Sonam Peldon.

**Software:** Leon B. Ellwein, Yuddha Dhoj Sapkota.

**Supervision:** Nor Tshering Lepcha, Gopal Prasad Pokharel, Taraprasad Das, Tandin Dorji, Sonam Peldon.

**Validation:** Indra Prasad Sharma, Nor Tshering Lepcha, Tshering Lhamo, Leon B. Ellwein, Gopal Prasad Pokharel, Yuddha Dhoj Sapkota, Tandin Dorji.

**Visualization:** Indra Prasad Sharma, Nor Tshering Lepcha, Tshering Lhamo, Gopal Prasad Pokharel, Tandin Dorji.

**Writing – original draft:** Indra Prasad Sharma, Leon B. Ellwein, Taraprasad Das.

**Writing – review & editing:** Indra Prasad Sharma, Nor Tshering Lepcha, Tshering Lhamo, Leon B. Ellwein, Gopal Prasad Pokharel, Taraprasad Das, Yuddha Dhoj Sapkota, Tandin Dorji, Sonam Peldon.

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
