## [Decision Letter · Decision Letter 0]

26 May 2020

PONE-D-20-11483

Visual impairment and refractive error in children in Bhutan: the findings from the Bhutan School Sight Survey (BSSS 2019)

PLOS ONE

Dear Dr. Sharma,

Thank you for submitting your manuscript to PLOS ONE. After careful consideration, we feel that the article has merit but some points have to be addressed before it can be considered further. Therefore, we invite you to submit a revised version of the manuscript that addresses the points raised during the review process.

We look forward to receiving your revised manuscript.

Kind regards,

Yu-Chi Liu, M.D

Academic Editor

PLOS ONE

Journal Requirements:

2. Please provide additional details regarding participant consent. In the ethics statement in the Methods and online submission information, please ensure that you have specified that a parent or guardian provided written informed consent for each of the minors enrolled in this study.

Additional Editor Comments (if provided):

Reviewers' comments:

Reviewer's Responses to Questions

**Comments to the Author**

1. Is the manuscript technically sound, and do the data support the conclusions?

Reviewer #1: Partly

Reviewer #2: Yes

2. Has the statistical analysis been performed appropriately and rigorously? 

Reviewer #1: Yes

Reviewer #2: Yes

3. Have the authors made all data underlying the findings in their manuscript fully available?

Reviewer #1: Yes

Reviewer #2: No

4. Is the manuscript presented in an intelligible fashion and written in standard English?

Reviewer #1: No

Reviewer #2: Yes

5. Review Comments to the Author

Reviewer #1: Overall the paper needs improvements throughout all sections. The introduction needs to highlight the publications on the topic and relevance of this study. The results and discussion are the sections that need major improvement, as the message it’s no clear. The sections need to be reformulated to emphasize the visual impairment based on best corrected VA and not under corrected VA. The authors can also highlight the importance of present VA to show that many children are not well corrected or need glasses. Looking at visual impairment using best corrected VA we can see that the number of visual impaired children is very low. The number of uncorrected refractive cases is high, and this is a different concept that needs to be explained. Another important concept that needs explanation is the use of “prevalence of myopia” determined with cycloplegia. This does not look the most adequate for this study and might be misinterpreted, as only a subset of children with VA of 6/12 or below were properly refracted with cycloplegia and not all children. There is not enough discussion on the results (e.g. How this prevalence compares with other Asian regions? What about future studies that need to be conducted?, etc.).

Reviewer #2: Visual impairment (VI) rate has yet been well reported in Bhutan. In this regard, this study provides good value-add.

#1. Why did the authors focus on uncorrected VI rather than presenting VI ? I would think presenting VI is more relevant/ important.

#2. To be more consistent with US definition of VI – I would suggest the authors to define VI as <6/12 rather than ≤6/12.

#3. The hyperopia definition of +2.00 D or more seems rather stringent, and may inadvertently underestimate the true prevalence of hyperopia. Why not define as +1.00D or even +0.50 dioptre ? Any reason for this ? Given that Bhutan is at higher altitude, is there a higher prevalence of hyperopia compared to other countries ?

#4. I believe this is one of the first report of such from Bhutan – it would be nice if the authors can discuss and compare these findings with other previous population-based studies from other countries/ regions. And from there – perhaps set the scope for future public health, interventional directions on visual impairment prevention for school children in Bhutan.

#5. It would be good if the authors can also present age-standardised prevalence – standardized to national population census (for that age group) if possible.

#6. In addition to the current Table 3 – perhaps it is also meaningful to evaluate factors associated with presenting visual impairment.

6. PLOS authors have the option to publish the peer review history of their article (what does this mean?). If published, this will include your full peer review and any attached files.

Reviewer #1: Yes: Carla Lanca

Reviewer #2: No

---

## [Author Response · Author response to Decision Letter 0]

5 Jun 2020

'Responses to Reviewers' attached as a separate file (.docx) in the 'Attach File' section of the online submission.

Please note that the line numbers mentioned in the 'Response to Reviewers' corresponds to lines numbers in the 'Manuscript with Track Changes'

Thank you

---

## [Decision Letter · Decision Letter 1]

3 Aug 2020

PONE-D-20-11483R1

Visual impairment and refractive error in school children in Bhutan: The findings from the Bhutan School Sight Survey (BSSS 2019)

PLOS ONE

Dear Dr. Sharma,

Thank you for submitting your manuscript to PLOS ONE. After careful consideration, we feel that the manuscript has been improved. However, the reviewer still has some concerns that need to be addressed. Therefore, we invite you to submit a revised version of the manuscript that addresses the points raised during the review process.

We look forward to receiving your revised manuscript.

Kind regards,

Yu-Chi Liu, M.D

Academic Editor

PLOS ONE

Reviewers' comments:

Reviewer's Responses to Questions

**Comments to the Author**

1. If the authors have adequately addressed your comments raised in a previous round of review and you feel that this manuscript is now acceptable for publication, you may indicate that here to bypass the “Comments to the Author” section, enter your conflict of interest statement in the “Confidential to Editor” section, and submit your "Accept" recommendation.

Reviewer #1: (No Response)

Reviewer #2: All comments have been addressed

2. Is the manuscript technically sound, and do the data support the conclusions?

Reviewer #1: Partly

Reviewer #2: Yes

3. Has the statistical analysis been performed appropriately and rigorously? 

Reviewer #1: Yes

Reviewer #2: Yes

4. Have the authors made all data underlying the findings in their manuscript fully available?

Reviewer #1: Yes

Reviewer #2: No

5. Is the manuscript presented in an intelligible fashion and written in standard English?

Reviewer #1: Yes

Reviewer #2: Yes

6. Review Comments to the Author

Reviewer #1: The paper is much improved with the revisions done by the authors. However, I still feel that visual impairment results should rely on best corrected VA and not under corrected VA. The prevalence of visual impairment using best corrected VA is very low (only 0.34%) and this finding is ignored in the discussion section. The paper would benefit from having two sections in the results, 1) highlight the importance of present VA (compared to uncorrected VA) to show that many children are not well corrected or need glasses; The number of uncorrected refractive cases is high. 2) Visual impairment based on best corrected VA.

Reviewer #2: All previous comments have been well addressed by the authors during this current revision.

This work will add good and new knowledge on the prevalence of visual impairment in Bhutan -- which has yet been widely reported and understood.

Thank you.

7. PLOS authors have the option to publish the peer review history of their article (what does this mean?). If published, this will include your full peer review and any attached files.

Reviewer #1: **Yes: **Carla Lanca

Reviewer #2: No

---

## [Author Response · Author response to Decision Letter 1]

10 Aug 2020

Reviewer #1: 

Reply from Authors: Thank you for your comments. Visual impairment results for uncorrected, presenting, and best corrected VA are presented in Table 1 in the Results Section. A new paragraph is added (Lines 335-342) as suggested by the reviewer.

Reviewer #2

Reply from Authors: Thank you.

---

## [Decision Letter · Decision Letter 2]

1 Sep 2020

Visual impairment and refractive error in school children in Bhutan: The findings from the Bhutan School Sight Survey (BSSS 2019)

PONE-D-20-11483R2

Dear Dr. Sharma,

We’re pleased to inform you that your manuscript has been judged scientifically suitable for publication and will be formally accepted for publication once it meets all outstanding technical requirements.

Kind regards,

Yu-Chi Liu, M.D

Academic Editor

PLOS ONE

Additional Editor Comments (optional):

Reviewers' comments:

Reviewer's Responses to Questions

**Comments to the Author**

1. If the authors have adequately addressed your comments raised in a previous round of review and you feel that this manuscript is now acceptable for publication, you may indicate that here to bypass the “Comments to the Author” section, enter your conflict of interest statement in the “Confidential to Editor” section, and submit your "Accept" recommendation.

Reviewer #1: All comments have been addressed

2. Is the manuscript technically sound, and do the data support the conclusions?

Reviewer #1: Yes

3. Has the statistical analysis been performed appropriately and rigorously? 

Reviewer #1: Yes

4. Have the authors made all data underlying the findings in their manuscript fully available?

Reviewer #1: Yes

5. Is the manuscript presented in an intelligible fashion and written in standard English?

Reviewer #1: Yes

6. Review Comments to the Author

Reviewer #1: (No Response)

7. PLOS authors have the option to publish the peer review history of their article (what does this mean?). If published, this will include your full peer review and any attached files.

Reviewer #1: **Yes: **Carla Lanca

---

## [Editor Report · Acceptance letter]

4 Sep 2020

PONE-D-20-11483R2 

Visual impairment and refractive error in school children in Bhutan: The findings from the Bhutan School Sight Survey (BSSS 2019) 

Dear Dr. Sharma:

I'm pleased to inform you that your manuscript has been deemed suitable for publication in PLOS ONE. Congratulations! Your manuscript is now with our production department. 

Kind regards, 

on behalf of

Dr. Yu-Chi Liu 

Academic Editor

PLOS ONE